# Building Knowledge Graphs from Survey Data: A Use Case in the Social Sciences

Lars Heling[1], Felix Bensmann[2], Benjamin Zapilko[2],
Maribel Acosta[1], and York Sure-Vetter[1]

[1] Institute AIFB, Karlsruhe Institute of Technology (KIT)
`firstname.lastname@kit.edu`
[2] GESIS - Leibniz Institute for the Social Sciences
`firstname.lastname@gesis.org`

**Abstract.** Many research endeavors in the social sciences rely on high-quality empirical data. Survey data is often used to investigate social and political behavior. The GESIS Panel is a probability-based mixed-mode panel survey in Germany providing high-quality survey and statistical data about e.g. political opinions, well-being, and other contemporary societal topics. In general, the process for integrating and analyzing the relevant data is very time-consuming for researchers. This is due to the fact, that search, discovery, and retrieval of the survey data require accessing various data sources providing different information in different file formats. In this paper, we present our architecture for building a Knowledge Graph of the GESIS Panel data. We present the relevant heterogeneous data sources and demonstrate how we semantically lift and interlink the data in a shared RDF model. At the core of our architecture is the Knowledge Graph representing all aspects of the surveys. It is generated in a modular fashion and therefore, our solution can be transferred to the existing infrastructure of other survey data publishers.

**Keywords:** Knowledge Graph, Survey Data, RDF, DDI

## 1 Introduction and Motivation

Linked Open Data initiatives have led to an increasing amount of data being published using the Resource Description Framework (RDF) on the web. At the core of RDF is the concept of linking resources within or across RDF graphs such that the resulting dataspace can be understood as a Knowledge Graph (KG) [7]. This allows data publishers to independently administer and publish their own data and improving its value and visibility by linking it to data of other publishers offering similar or additional information on the resources. In this paper, we present an in-use application of such a KG in the domain of the social sciences at GESIS - Leibniz Institute for the Social Sciences. Our work is motivated by the circumstance that data related to the GESIS Panel[3] like questionnaires and observation data is administered and published in different

---

[3] `https://www.gesis.org/en/gesis-panel/gesis-panel-home/`

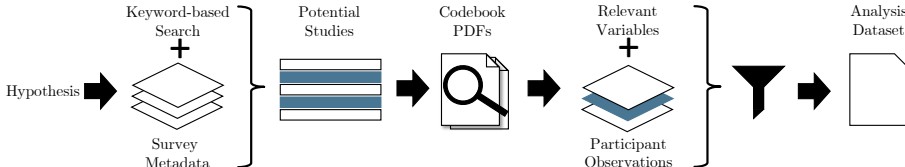

Fig. 1: Motivation: Current process to retrieve survey data based on a hypothesis.

datasets varying in format and representation. As a result, the current process for researchers aiming to use the rich collection of surveys available at GESIS requires manually consulting different information sources to discover and obtain relevant data which is a time-consuming task.

**Motivating Scenario.** Consider the current process to discover and retrieve the data from the GESIS Panel outlined in Figure 1. A researcher has a research question and formulates a hypothesis according to which she aims to investigate by leveraging the data provided by the GESIS Panel. Typically, the researcher first starts to discover the available survey datasets by a keyword-based search in the Data Catalog (DBK)[4], which is the online portal to search and retrieve survey related data. The search results are a list of surveys which match the keyword on the survey-level metadata, e.g. in the abstract summarizing the survey. Based on this list, the researcher can retrieve the codebook PDFs for all surveys from the portal. In the codebooks, the variables assessed in the surveys are detailed, and the researcher may search for all relevant variables. To obtain the final analysis dataset, the researcher needs to access the CSV documents with the recorded participant answers (or observations) for the relevant variables. In some cases, a download from the DBK is not available because of data protection laws and researchers are required to physically visit the Secure Data Center Safe Room at GESIS to access and work with the data on-site. After retrieving the final dataset, the researcher may use statistical analysis tools to investigate the hypothesis. This tedious process from a hypothesis to gaining first insights into the actual data impedes the research process for social scientists.

The goal of building a KG for the survey data is improving this process for researchers by facilitating the discovery and retrieval of relevant data. Using Semantic Web technologies as a foundation allows for publishing and linking data of independent sources providing a holistic picture of the GESIS Panel in the form a KG. Therefore, the contributions of this work are the following:

**C1** Description and analysis of a real world scenario from the social sciences domain with corresponding requirements,

**C2** Outline of our solution to handle data organization requirements by applying Semantic Web technologies to create a Knowledge Graph, and

**C3** Presentation of encountered challenges, lessons learned, and indication of future extensions.

---

[4]`https://dbk.gesis.org/dbksearch/`

In addition, we provide a demo[5] allowing access to parts of the KG. The remainder of this paper is structured as follows. In Section 2, we provide the preliminaries by introducing the GESIS Panel and relevant vocabularies, i.e., the DDI and the DDI-RDF Discovery Vocabulary. In Section 3, we present the architecture of our approach. We then revisit our motivating scenario and outline challenges encountered and lessons learned in Section 4 and analyze related work in Section 5. We summarize our work in Section 6 and indicate future works.

## 2     Preliminaries

In the following, we introduce the GESIS Panel, the Data Documentation Initiative (DDI) and the corresponding DDI-RDF Discovery Vocabulary.

### 2.1    GESIS Panel

The GESIS Panel[3] is a probability-based mixed-mode panel survey in Germany which is open to the research community [3]. The goal is obtaining high-quality survey data by employing a cross-sectional or longitudinal survey design. Probability-based indicates a participant selection optimized to accurately estimating the target population, which are German-speaking persons between age 18 and 70 who live in private households in Germany. Mixed-mode refers to the two modes of the data collection process, namely via web-based surveys or via traditional paper-and-pencil surveys sent to the participants. The data collection is performed periodically in *waves* on a bimonthly basis with a new questionnaire in each period, producing a continuously growing dataset. The data is published in three editions: standard edition, extended edition and campus file, each covering different subsets of the recorded data. Standard edition and campus file can be retrieved online, while the extended edition may only be accessed within the aforementioned Safe Room. The data collected in the GESIS Panel may serve as a basis for analyses in the social sciences and it has been used in several studies, for example, to examine the political opinions of the German population [4,6].

### 2.2    DDI and DDI-RDF Discovery Vocabulary

The Data Documentation Initiative (DDI)[6] is an internationally acknowledged standard to facilitate data management by documenting metadata on the datasets in the area of social, behavioral and economic sciences [9]. Therefore, the standard aims to improve data quality and ensure the long-term preservation of the information and it is driven by an alliance of data producers, archivists and users to jointly collaborate on the standard [9]. The DDI-RDF Discovery Vocabulary[7] (`disco`) aims at transferring the DDI standard to the Linked Data community.

---

[5]`https://km.aifb.kit.edu/services/gesispanel/demo`

[6]`https://www.ddialliance.org`

[7]`http://rdf-vocabulary.ddialliance.org/discovery.html`

It is based on a subset of DDI allowing for describing survey data in the social sciences which facilitates the discovery of this data and related metadata [1,2]. At the core of the vocabulary is the `Study` class which represents the generation process of a dataset. A set of studies is compiled in a `StudyGroup` in case the surveys are conducted in a continuous or periodic process. For example, each wave of the GESIS Panel can be modeled as a `Study` and they are combined into one `StudyGroup`. The content of the physical dataset holding the actual original survey data is represented in a `LogicalDataSet` for which licensing information and access policies may be attached. The content of a dataset is described by `Variable`s. Variables represent different aspects which are measured as part of a `Study` and, thus, are typically the columns in a tabular representation of the survey records. The data of a survey is commonly collected using a `Questionnaire` which consists of a set of `Question`s to measure the variables. Variables are associated with a `Representation` which is typically the set of answers for the associated question and the corresponding notation used in the dataset. The `Representation` is linked as the `responseDomain` to a question. Furthermore, the target population of a `Study` may be described using the classes `Universe` and `AnalysisUnit`. For instance, the target population of the GESIS Panel is a representative sample of the German population and, thus, the analysis unit is persons. The development of an RDF vocabulary along with the already existing DDI standard is motivated by various use cases which mostly support the discoverability of the data [1,11]. For instance, free text keyword-based search may be enabled and once studies and relevant data has been found, related studies and additional data may be discovered exploiting the links across the datasets.

## 3    Building a Knowledge Graph for the GESIS Panel

The goal of building a Knowledge Graph (KG) for the GESIS Panel by semantically lifting the original data sources to a shared RDF data model is improving the discovery, search and retrieval of survey data for social scientists. In the following, we provide an overview of the architecture and thereafter, describe the original data sources as well as the semantic lifting process in more detail.

### 3.1    Architecture

Figure 2 provides an overview of our architecture and the main components. From an integration perspective, the integration process is visualized in a bottom-up manner. At the bottom are the data sources providing different parts of data associated with the GESIS Panel:*i*) the access right management data associated with the datasets, *ii*) the survey metadata providing general information about surveys and corresponding waves, *iii*) the codebooks with information on how the variables in a survey are to be interpreted, and *iv*) the participant observations (unit-records) which encode the respondents' answers to the questionnaires. The data sources vary in format and schema. Therefore, each data source requires a custom *semantic lifting process* to transfer the original data to

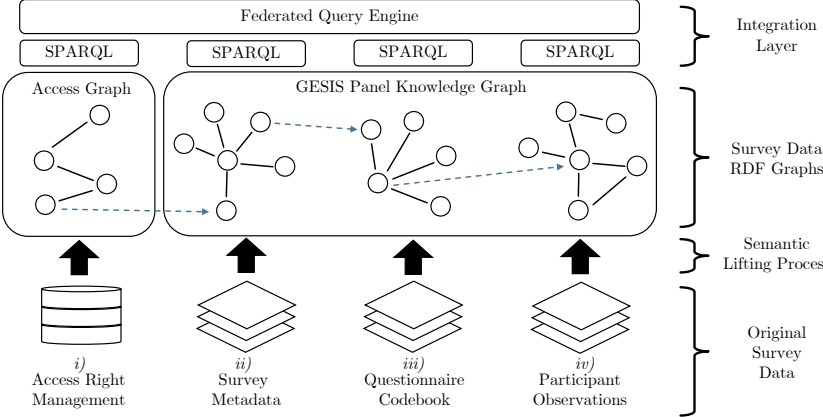

Fig. 2: Architecture of the infrastructure for building the GESIS Panel KG.

the shared RDF data model. Since the data sources are heterogeneous in nature and their maintenance is deeply rooted in and grown together with the organization of GESIS, the lifting processes need to be invoked individually, whenever updates on a specific dataset are to be made. Each semantic lifting process takes an original data set as input and returns an RDF graph. By defining conventions for naming resources (URIs) of common instances across the different data sources, they are interlinked across the RDF graphs. As a result, each graph stands for itself but combined together they provide a holistic KG of the GESIS Panel. In our implementation, each graph is provided via an individual SPARQL endpoint as this allows for the original data providers to independently manage and publish their data. Furthermore, survey metadata and codebook data may be offered via public endpoints to allow researchers to discover available data while parts of the participant observations may only be accessed within the Safe Room to comply to data security and privacy regulations of GESIS. Finally, at the top is the integration layer consisting of SPARQL endpoints which may be accessed by a federated engine to query the KG. The integration layer may be directly queried by users or, alternatively, accessed by an application such as a GUI. In our specific case, the non-sensitive data may be merged and provided by a single endpoint for performance improvements when querying the data. Here we present the generic architecture as this may not be applicable in any organization. According to this bottom-up architecture and existing processes at GESIS, changes in individual data sources are propagated from the original data to the KG via a semantic lifting process.

## 3.2   GESIS Panel Knowledge Graph

We provide a simplified example extract of the GESIS Panel KG in Figure 3 to exemplifying the RDF graphs from the different data sources and how they are interlinked. Thereafter, we detail the semantic lifting process. Starting at

the top, Figure 3a shows user `User1` and a `RightStatement` to indicate the user's permissions to access the extended edition of GESIS Panel data. This `RightStatement` is linked to the metadata for the survey `WaveA` as shown in Figure 3b. The figure shows a subset of the original metadata that includes the `LogicalDataSet` providing the data, title, subjects, variables of the wave, as well as the time period in which the wave was conducted. In the example, merely the variable *Gender* is shown for the wave. The variable is linked to the questionnaire codebook subgraph shown in Figure 3c which provides details for the variable such as the question text and corresponding answers as well as their notation. Moreover, each variable has a corresponding property which is used in the participant observation subgraph. Figure 3d shows the recorded data for a participant represented using the RDF Data Cube Vocabulary[8]. Each observation is a blank node linked to the participant's identifier and the recorded variable values using the corresponding properties. In the example, the participant is *male* according to the notation provided in the questionnaire codebook.

### 3.3   Original Data and Semantic Lifting

In the following, we describe the original data available at GESIS and detail how the data is semantically lifted to the shared RDF data model of our KG.
**Access Right Management.** Considering the access policies, there are three different editions of the GESIS Panel: campus file, standard edition, and extended edition. In each edition, a different subset of survey data and variables are available. Accordingly, the access rights need to be defined on this level. The Dublin Core vocabulary (`dcterms`), which is reused in the `disco` vocabulary, allows for defining such access right statements on the level `LogicalDataSet`s and the data is associated with the corresponding access rights according to the edition. Furthermore, a user model is employed to define users and link them to the access right statements. Currently, this process is implemented in a manual fashion, however, we aim to integrate the access right management for our KG to existing solutions, such as the Lighweight Direcory Access Protocol (LDAP).
**Survey Metadata.** The Data Catalog (DBK)[4] is the online portal provided by GESIS to search and retrieve survey data including the GESIS Panel. The DBK operates on metadata describing surveys as a whole but not on the level of individual variables. Important aspects in the metadata are, for instance, citation data, version information, date of collection, or methodology. The survey-level metadata may also be retrieved from an internal database as XML documents following the DDI standard, where the data is continuously updated in an automatic fashion. As the GESIS Panel is considered as a single evergrowing survey, it is represented in a single large DDI file comprising the information of all associated waves. However, each time a wave is added, a new version is created for researchers to keep track of data provenance. We choose to represent the survey on the level of waves as individual `Study`s to allow for a consistent and retraceable mapping to the corresponding concepts of the `disco` vocabulary.

---

[8]`https://www.w3.org/TR/vocab-data-cube/`

(a) Access Right Management

(b) Survey Metadata

(c) Questionnaire Codebook

(d) Participant Observation

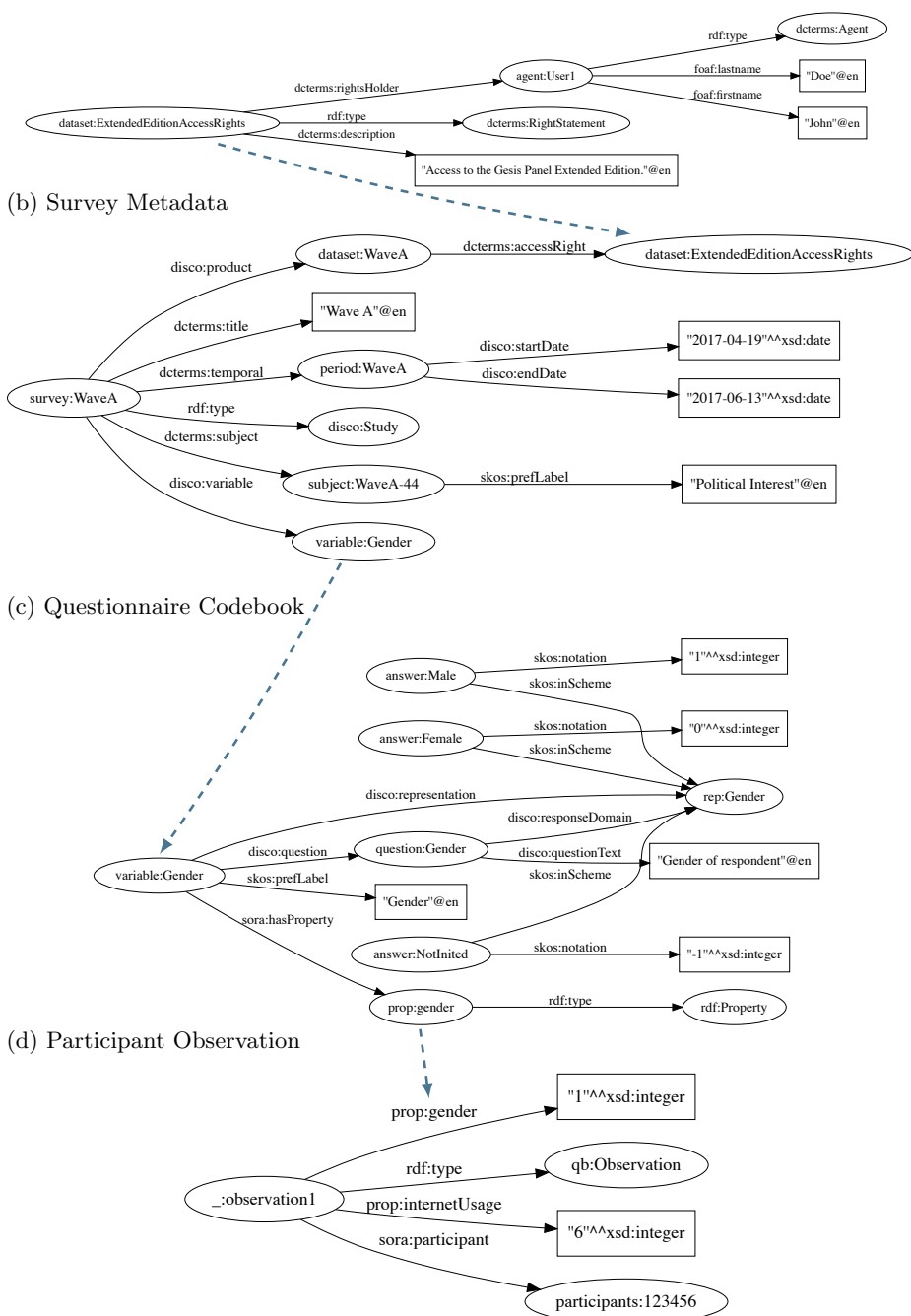

Fig. 3: Knowledge Graph Extract: The figures visualize the subgraphs of the Knowledge Graph to provide an overview of the shared RDF data model and the interlinking between the data sources. The dashed arrows indicate these relationships between shared resources. (The prefixes for `dcterms`, `disco`, `foaf`, `rdf`, `skos` and `qb` are used as in `prefix.cc`. The `sora` prefix is used for our vocabulary[9]. The other prefixes adhere to the scheme `http://.../gesis/resource/<prefix>/`.)

Table 1: Questionnaire Codebook: Simplified example extract of the codebook for a variable assessing a participant's gender measured in two surveys.

| varname | labelEn | questionText | code | valueLabel | waveID | betweenCorrespondence | ... |
|---|---|---|---|---|---|---|---|
| gender | Gender | Gender of the respondent | -1 | Not Inited | WaveA | genderB | ... |
| gender | Gender | Gender of the respondent | 0 | Female | WaveA | genderB | ... |
| gender | Gender | Gender of the respondent | 1 | Male | WaveA | genderB | ... |
| ... | ... | ... | ... | ... | ... | ... | ... |
| genderB | Gender | Gender of the respondent | -1 | Not Inited | WaveB | | ... |
| genderB | Gender | Gender of the respondent | 0 | Female | WaveB | | ... |
| genderB | Gender | Gender of the respondent | 1 | Male | WaveB | | ... |
| ... | ... | ... | ... | ... | ... | ... | ... |

**Questionnaire Codebook.** The detailed information on the variables and the corresponding questionnaires for each wave are provided in *codebooks*. For researchers, the codebooks are accessible as PDF files in the DBK portal. Internally, these PDFs are generated from CSV files containing all the necessary information in a tabular format. The data for the example extract is shown in Table 1. Variables are uniquely identified by a `varname` which is based on the identifier of the corresponding wave (i.e., `waveID`) and a number (omitted for visualization purposes). Variables are represented in the rows and depending on the type of question and set of answers, several rows represent a single variable. This type of tabular representation requires two major aspects to be considered in the semantic lifting process:*i*) information related to variables such as corresponding answers and their notation may be stored in a redundant fashion, and *ii*) semantically identical variables are assessed in various waves but differently identified in each wave, which requires means of linking them. In order to address the first aspect, the URI for the representation of a variable is derived from the URI of the variable itself, e.g. `variable:Gender` → `rep:Gender`. The URIs for answers are based on hashing the German and English answer text in combination with the notation of the answer. Consequently, reoccurring answers are identified by existing URIs to avoid redundancies. The second aspect is addressed by defining properties as part of our own vocabulary[9], prefixed with `sora`, to link the reoccurring variables. For instance, `sora:betweenCorrespondence` links a variable to a similar variable in another survey. The vocabulary also provides terms to associate variables with additional information from the original CSV files, such as introductory texts for the questions.

**Participant Observations.** The last data source provides the participant observations, i.e., the answer provided by the participants to the questions of each survey. The observations are provided in a tabular form with each row corresponding to a participant and the columns to the measured value for the variables. The column names for the variables are the `varname` identifiers from the codebooks. The data is distributed over several CSV files which allows restricting the access according to the aforementioned editions. The first columns provide

---

[9]`https://w3id.org/sora/resource/vocabulary`

basic metadata, such as the unique personal identifier (`PID`) of the participants and the version of the data. In the semantic lifting process, for each row, a data cube `Observation` is instantiated as a blank node and it is linked to the respective participant URI which is derived from the `PID`. This allows linking the observations for the same participant across several studies. The answers are added to the observation using the corresponding property. Since the identifiers `varname` from the codebooks and the column names in the observation files coincide, the URIs for the properties can be created independently of the codebooks. For example, column *gender* → `prop:gender`.

**Semantic Lifting.** A major requirement for the semantic lifting process was a lightweight solution applicable to all data sources to facilitate maintenance and adoption. We investigated existing mapping languages and integration tools, coming to the conclusion that due to discrepancies in the structure of some original data sources and the target DDI RDF data model, there is no single out-of-the-box solution for our use-case. For instance, as in the DDI XML standard, the GESIS Panel is considered a single (evergrowing) study, we need to apply complex regular expressions on the textual study description containing the information on all waves to generate the RDF data. As a result, the data cleaning, semantic lifting, and refinement process is currently implemented in Jupyter notebooks[10] using Python with RDFLib[11]. In the future, we aim to further investigate mapping approaches to improve the lifting process with a solution suitable for both the partially unstructured and statistical data.

## 4   Challenges and Lessons Learned

In the following, we revisit the motivating scenario and show how the discovery and retrieval of survey data is facilitated by our solution. Thereafter, we discuss the challenges we encountered and present our solutions to overcome them.

In Section 1, we outlined the current process for a researcher to follow in order to search, discover, and retrieve survey data of the GESIS Panel. As our example shows, the current process requires consulting various data sources in different formats and representations entailing a very time-consuming process for users. The shortcoming is addressed by our approach as the KG may be queried via a single interface accessing the data represented in the shared RDF model. As a result, federated SPARQL queries may be processed over the different data sources which improve both the discovery of relevant data as well as its retrieval. Moreover, a more fine-grained search leveraging the information of the questionnaires is supported as this information is represented in our KG. Let us consider the following example: "Find the questions and surveys where the variable label or the question text contains the term *politic* from the Extended Edition of the GESIS Panel of surveys conducted in 2014". In the current process, this would require searching all surveys conducted in 2014 and examining every corresponding codebook PDF manually for the term *politic*. In contrast, the

---

[10] https://jupyter.org/
[11] https://github.com/RDFLib

Listing 1.1: SPARQL query exemplifying a search to retrieve questions and corresponding waves related concerning politics.

```
0  SELECT ?question ?wave WHERE {
1     # Retrieve accessible waves
2     ?wave disco:product [ dcterms:accessRights dataset:ExtendedEditionAccessRights ] .
3     ?wave dcterms:temporal [disco:endDate ?end ; disco:startDate ?start ] .
4     # Filter according to the time period
5     FILTER (?start >=''2014−01−01''^^xsd:date && ?end <''2015−01−01''^^xsd:date)
6     # Retrieve variable label and question text
7     ?wave disco:variable ?variable .
8     ?variable skos:prefLabel ?varLabel .
9     ?variable disco:question [ disco:questionText ?question ].
10    # Filter according to the keyword
11    FILTER (regex(?varLabel, ''politic'')  regex(?question, ''politic''))  }
```

required information can be retrieved from our KG using a single SPARQL query executed by a federated query engine. The query is shown in Listing 1.1. In lines 2 and 3, the information for the survey are retrieved and filtered in line 5 to include only surveys from 2014. In line 7, all associated variables are selected and the corresponding label and question texts are selected in lines 8 and 9. Finally, the results are filtered in line 11 according to the keyword *politic*.

Several challenges need to be addressed when building the KG which mainly originate from the organizational structure of GESIS as well as data security and privacy requirements. The relevant GESIS Panel data is distributed across and administered in different datasets with varying formats and schema, hindering merging the data in a single repository. We address by defining semantic lifting processes for each data source to create multiple RDF graphs. By defining conventions for naming resources (i.e., the URIs) the resulting graphs are interlinked and can be understood as single KG. Whenever a new version of the graph is created, the data is published via the corresponding SPARQL endpoints such that a federated query engine may process queries across the entire KG. Furthermore, this semi-automatic and curated semantic lifting process to generate the RDF graphs were chosen to assure the provision of high-quality data. Sensitive data may be provided by endpoints only accessible in the network of the Safe Room at GESIS which addresses the data security and privacy requirements.

Concluding, the major lesson learned was embracing the existing infrastructures at the organization and develop the architecture accordingly. This will lead to a sustainable solution which is easier to maintain in the future. Moreover, we learned that the `disco` vocabulary is suitable for our use case as it covers almost all relevant aspects of our KG. Defining additional terms to cover all aspects specific to the data publisher is inevitable.

## 5    Related Work

The application of Semantic Web technologies and publishing data according to the Linked Data principles have been studied in the area of social sciences.

The motivation of most research endeavors is improving the discovery and use of survey and statistical data by publishing metadata and facilitating integration.

Similar to our work, Gottron et al. [5] address the problem of survey and statistical data being scattered across various files and data sources. They propose the use of open semantic models to facilitate the searching, merging and aggregating such distributed data. They present the Semantic Digital Library of Linked Data, a framework tailored for the social sciences. The authors mainly focus on integrating and merging data from two data sources, survey and statistical data, on an aggregation level using the Data Cube vocabulary[8]. In contrast to our work, they do not include search capabilities in their prototype but mostly focus on visualization and lightweight calculations over the integrated data.

Zapilko et. al [10] present a solution to provide a linked thesaurus for the social sciences (TheSoz), which is essential for indexing documents and research information in the social sciences, such as survey descriptions. The original thesaurus is administered in a database and the authors present their approach transforming the thesaurus to Linked Data using the Simple Knowledge Organization System (SKOS)[12] standard. The presented thesaurus allows for connecting heterogeneous datasets and therefore, may be linked to our KG to further improve the discovery of survey data by leveraging annotations using TheSoz.

Similarly, Schaible et al. [8] aim to interlink study descriptions to the Linked Open Data Cloud. They also transform DDI XML documents providing the study-level descriptions to RDF in order to link the resulting dataset to resources from the Integrated Name Authority File (GND) and DBpedia. For the linking task, they use Silk[13] to discover and generate `owl:sameAs` links between a source dataset and target datasets. In contrast to our work, the authors focus on the study-level metadata and do not provide a holistic approach spanning all relevant data sources and the associated intricacy, such as privacy requirements. Furthermore, their primary goal was linking resources from the description to other resources in the LOD Cloud. As a result, their approach may be applied as a subsequent refinement step after the KG is created and may be extended to the semantic data about variables and questionnaires produced with our solution.

## 6   Conclusions

In this work, we presented our solution to build a Knowledge Graph (KG) for survey data to facilitate search, discovery, and retrieval of survey data. This is achieved by semantically lifting data from heterogeneous data sources to a shared RDF data model based on the DDI-RDF Discovery Vocabulary and providing the result graphs via SPARQL endpoints to enable the integration using federated SPARQL query processing while adhering to data security and privacy requirements. The presented solution overcomes various challenges and requirements common to organizations publishing survey data and, therefore, may be applied in other organizations as well. The described architecture is used in the

---

[12]`https://www.w3.org/2004/02/skos/`
[13]`http://silkframework.org/`

$SoRa$[14] project which aims to link survey data with geospatial information. In the future, we aim at extending our current approach by including an additional refinement step to further enhance the information in the KG. For example, similar to [8], NLP may be applied to extract information from the questions text and survey descriptions to associate them with specific topics or link them to named entities. In addition, we want to extend the KG to include further surveys conducted by GESIS. Furthermore, as part of the $SoRa$ project we will apply our solution at other survey data providers, namely the German Socio-Economic Panel[15]. Ultimately, future work may focus on enabling querying KGs of survey data across different organizations.

### Acknowledgments

This work was carried out with the support of the German Research Foundation (DFG) within the project "Sozial-Raumwissenschaftliche Forschungsdateninfrastruktur"[14].

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
