# OpenReview forum: "Building Knowledge Graphs from Survey Data: A Use Case in the Social Sciences"
_eswc-conferences.org/ESWC/2019/Workshop/KGB — KGB 2019_

### Official Review · ~David_Chaves-Fraga1 · 2019-03-27
**Lack of standard processes for KG Building**

**Rating:** 3
**Confidence:** 3

**Review:**

In this paper, a use case about the generation of a Knowledge Graph from Survey Data. The authors use a German platform that contains surveys and statistical data about social science and generate the corresponding knowledge. The paper is well written, easy to follow and with good motivation. I like so much the given value of having a KG instead of raw data that is reflected in the paper.

My main concern about this paper are the semantify process of the data sources. The authors explain that the platform provides 4 different sources (survey metadata, questionnaire codebook, participant observations and access right management) and they propose an individual approach for each data source to generate the corresponding KG. The issues here are that they don't explain anything about how the lifting process is performed or what technologies they are using (only in the case of the access right management they say that is a manual integration). For example (section 3.3 - survey data): "the data for each wave is extracted from the XML document and then the Study resources with the corresponding metadata are created" okey, but how? Do you use mappings? RML? Other? Do you use openrefine/karma?. I would strongly recommend rewriting section three, be focused on what technologies are used and how the KGs are generated.  Following a standard methodology with common technologies (R2RML, RML) will ensure the reproducibility of the KG process creation and will also give robustness to it. If the current technologies are not enough for your proposal, at least say why and motivate your lifting processes, but an explanation is needed. Now, this section is basically defining the vocabulary (as I understood is a standard for this domain) and relations among the different KGs that can be summarized in the previous section.

About the architecture of the proposal,  it is not clear to me the main motivation of having a federated query engine in the top, is not enough one SPARQL endpoint with the 4 different graphs identifiers? If the problem is managing the access to the data sources, 2 KGs could be enough and the performance of the queries may be improved.

In the end, I don't understand why the focus of the related work is linked data and social sciences but anything about KG building methods.  For example, there are methods that deal with the generation of virtual knowledge graph (e.g., proposing SPARQL-to-SQL techniques) for always obtain updated SPARQL results [1,2]. Assuming that a lot of the data source are statistical data the authors should also check [3] (please, contact me if you have any question).

 As the paper is well motivated and also is a very good use case of having a KG instead of raw data. I would like to see it in the workshop but the issues I mentioned should be resolved for the camera ready version.

[1] Priyatna, F., Corcho, O., & Sequeda, J. (2014, April). Formalisation and experiences of R2RML-based SPARQL to SQL query translation using morph. In Proceedings of the 23rd international conference on World wide web (pp. 479-490). ACM.
[2] Calvanese, D., Cogrel, B., Komla-Ebri, S., Kontchakov, R., Lanti, D., Rezk, M., ... & Xiao, G. (2017). Ontop: Answering SPARQL queries over relational databases. Semantic Web, 8(3), 471-487.
[3] Chaves-Fraga,  D.,  Priyatna,  F.,  Perez-Santana,  I.,  Corcho,  O.:  Virtual statistics knowledge graph generation from CSV files. In: Emerging Topics in Semantic Technologies: ISWC 2018 Satellite Events. Studies on the Semantic Web, vol. 36, pp.235–244. IOS Press (2018). Available at: http://dchaves.oeg-upm.net/resources/papers/virtual-semstat-2018/

---

### Official Review · ~Anastasia_Dimou1 · 2019-04-06
**Knowledge Graph Building for survey data from Social Sciences**

**Rating:** 4
**Confidence:** 3

**Review:**

This paper presents how knowledge graphs are built from survey data in the field of social sciences. To be more precise, it is presented how the knowledge graph of the GESIS panel is built. The data in GESIS panel are collected with web-based or paper-and-pencil surveys. Therefore, the available data consists of survey data accompanied by the relevant questionnaires and participant observations, as well as the corresponding access rights for the different sets of data. The data is published in three editions: standard, extended and campus.

The paper is very well written, it clearly describes the current situation, the limitations caused to the users who try to access this data and the potential that semantic web technologies bring. The high level architecture is well presented, however details regarding the actual knowledge graph building, ie the semantic lifting, are not described. Considering that this is the focus of the workshop, it would be interesting to have some more details about the requirements and limitations and, thus, how the solution was designed, how the used tools were chosen and which challenges were addressed. Which approaches/tools were investigated as candidates, which were the limitations and what determined which tool was chosen? Were the available tools sufficient for the needs and, if not, what were they lacking of? I would suggest such details to be included in the paper and discussed during the presentation. It would be very interesting for the audience and for raising discussions.

---

### Decision · Program_Chairs · 2019-04-08
**Acceptance Decision**

**Decision:**

Accept

**Comment:**

This contribution is accepted for presentation at the KGB2019 workshop, and for inclusion in its proceedings.